# Isolation of TTF-1 Positive Circulating Tumor Cells for Single-Cell Sequencing by Using an Automatic Platform Based on Microfluidic Devices

**DOI:** 10.3390/ijms232315139

**Published:** 2022-12-01

**Authors:** Hei-Jen Jou, Hsin-Cheng Ho, Kuan-Yeh Huang, Chen-Yang Chen, Sheng-Wen Chen, Pei-Hsuan Lo, Pin-Wen Huang, Chung-Er Huang, Ming Chen

**Affiliations:** 1Departments of Obstetrics and Gynecology, Taiwan Adventist Hospital, Taipei 105404, Taiwan; 2Departments of Obstetrics and Gynecology, National Taiwan University Hospital, Taipei 100225, Taiwan; 3School of Nursing, National Taipei University of Nursing and Health Science, Taipei 112303, Taiwan; 4Cytoaurora Biotechnologies Inc., Hsinchu Science Park, Hsinchu 302058, Taiwan; 5Department of Electrical Engineering, National Chung Cheng University, Chiayi 621301, Taiwan; 6Department of Emergency Medicine, Show Chwan Memorial Hospital, Changhua 500009, Taiwan; 7Department of Genomic Medicine, Changhua Christian Hospital, Changhua 500209, Taiwan; 8Department of Obstetrics and Gynecology, Changhua Christian Hospital, Changhua 500209, Taiwan; 9Department of Molecular Biotechnology, Da-Yeh University, Changhua 515006, Taiwan; 10Department of Obstetrics and Gynecology, College of Medicine, National Taiwan University, Taipei 106319, Taiwan; 11Department of Medical Genetics, National Taiwan University Hospital, Taipei 100225, Taiwan

**Keywords:** circulating tumor cells, single-cell sequencing, TTF-1^+^ CTCs, lung cancer, nanostructured microfluidic chip, automatic cell picker, EGFR, ERBB2, KRAS, STK 11

## Abstract

Single-cell sequencing provides promising information in tumor evolution and heterogeneity. Even with the recent advances in circulating tumor cell (CTC) technologies, it remains a big challenge to precisely and effectively isolate CTCs for downstream analysis. The Cell Reveal^TM^ system integrates an automatic CTC enrichment and staining machine, an AI-assisted automatic CTC scanning and identification system, and an automatic cell picking machine for CTC isolation. H1975 cell line was used for the spiking test. The identification of CTCs and the isolation of target CTCs for genetic sequencing were performed from the peripheral blood of three cancer patients, including two with lung cancer and one with both lung cancer and thyroid cancer. The spiking test revealed a mean recovery rate of 81.81% even with extremely low spiking cell counts with a linear relationship between the spiked cell counts and the recovered cell counts (Y = 0.7241 × X + 19.76, R^2^ = 0.9984). The three cancer patients had significantly higher TTF-1^+^ CTCs than healthy volunteers. All target CTCs were successfully isolated by the Cell Picker machine for a subsequent genetic analysis. Six tumor-associated mutations in four genes were detected. The present study reveals the Cell Reveal^TM^ platform can precisely identify and isolate target CTCs and then successfully perform single-cell sequencing by using commercially available genetic devices.

## 1. Introduction

Cancer is the leading cause of death worldwide with approximately 10 million cancer deaths or one-sixth of all deaths in 2020 [1]. It has been estimated that metastasis is responsible for as much as 90% of cancer-associated deaths [2]. Currently, the detection and characterization of circulating tumor cells (CTCs) may provide crucial information, which can assist early detection of the tumor, assessment of the prognosis, therapeutic response, and the detection of minimal residual disease [3,4,5]. Therefore, the detection and analysis of CTCs may provide an opportunity to take personalized cancer treatment a step forward.

CTCs are rare cells that shed from primary or metastatic tumor sites and then enter into the peripheral blood of cancer patients [6]. They play a crucial role in tumor metastasis and may represent tumor heterogeneity. Therefore, CTCs can be used as a “real-time liquid biopsy” to monitor tumor progression. However, there are only about 1–10 CTCs per mL of peripheral blood in patients with metastatic cancer [7]. Because of the rarity of CTCs, they need to be enriched and isolated from other blood cells before further analysis [8,9].

Recent advances in microfluidic techniques and automation of the laboratory procedures not only support the development of high-throughput CTC tests with more consistent results, but also enable further downstream single-cell analyses [10]. Research on single-cell analyses can help researchers to understand the phenotypes and characteristics of CTCs at single-cell level and help the physicians realize cellular heterogeneity as well the mechanism of tumor evolution and treatment resistance, which is helpful in the decision-making in cancer treatment [11].

The purpose of this proof-of-concept study includes the following: 1. To examine whether this system can efficiently isolate CTCs with specific markers for single-cell sequencing; 2. To preliminary verify whether TTF-1^+^ CTCs can be used as a marker for lung adenocarcinoma; 3. To examine tumor-associated mutations in TTF-1^+^ CTCs.

## 2. Results

### 2.1. Recovery Rate and Linearity of Cell Reveal^TM^ System

Different numbers (1419, 561, 237, 120, 29) of H1975 cells were spiked into PBMCs from healthy donor (isolated from 7.5 mL of blood). The number of cells spiked into PBMCs were plotted against cells detected by Cell Reveal^TM^. The slope, intercept, and correlation coefficient (R^2^) of the detected H1975 cells versus spiked H1975 cells were analyzed by linear regression (Figure 1). The average recovery rate of H1975 spiked into PBMCs was 81.81%. The result indicated that Cell Reveal^TM^ recovered the H1975 cells efficiently even with extremely low spiked cell accounts.

### 2.2. Identification and Isolation of CTCs

Table 1 lists the clinical diagnosis, cancer staging and results CTC test of the three patients with cancer. In addition to TTF-1, various tumor-associated markers were tested. The TTF-1^+^ CTC counts in peripheral blood were 13/4, 8/2, and 16/8 mL in cases 1, 2, and 3, respectively. There were 12 TTF-1^+^ CTCs in three CTC clusters (Figure 2) in case 3. On average, 5.3 (range: 3.5–8) CTCs and 3.2 (range: 2–4) TTF-1^+^ CTCs could be captured from per mL of blood, respectively. We attempted to isolate TTF-1^+^ CTCs isolated from 3 blood samples for genetic analysis. Such attempts were successful in both case 2 and case 3. Only six TTF-1^+^ CTCs were obtained in case 1. The other TTF-1^+^ CTCs were located in difficult-to-manipulate areas, such as the edge of the microfluidic channel. For avoiding damage to the microcapillary pipettes and target cells, TTF-1^−^ CTCs were therefore isolated to achieve sufficient cell numbers for genetic analysis. The purity for isolated CTCs ranged between 37 and 80%.

Figure 3 plots the TTF-1^+^ CTC count per mL of peripheral blood in the 10 healthy volunteers and three patients with cancer. Mean TTF-1^+^ CTC counts in healthy volunteers and the three cancer patients were 0.1/mL (ranging: 0–0.5/mL, SD: 0.21) and 3.2/mL (ranging: 2.0–4.0/mL, SD: 1.0), respectively. The difference of TTF-1^+^ CTC counts between the cancer patients and healthy volunteers was statistically significant (*p* < 0.05).

### 2.3. Average Sequencing Depth and On-Target Percentage

All CTC samples comprise 10 cells on average. The sequencing depth in the average of all samples is about 7142. The sequencing depth greater than 100× is sufficient for variant detection in general. An average on-target percentage (>10×) is 80% and all samples are greater than 60%. Even though the sequencing depth is greater than 250×, the on-target percentage is still greater than 50% (Table 2).

### 2.4. Mutations in Isolated CTCs

Table 3 summarizes the results of genetic sequencing. There were six mutations in four genes in these three patients. The mutation occurred in EGFR, ERBB2, KRAS, and STK11 genes (Figure 4).

## 3. Discussion

In the present proof-of-concept study, CTCs with tumor-associated markers were successfully identified and isolated by using an automatic Cell Reveal^TM^ platform and followed by genetic sequencing using commercially available devices. The Cell Reveal^TM^ system integrates nano-structured microfluidic chips, an automated CTC enrichment system, an AI-assisted cell identification and localization system, and an automated cell picking system. The Cell Reveal^TM^ platform can not only perform CTC profiling but can also accurately isolate single CTC with minimal cell loss, allowing to gather enough CTCs with specific biomarker for genetic sequencing. A multi-omics analysis of CTCs explores tumor evolution in cancer patients and therefore can serve as a “real-time liquid biopsy”. 

Because of the scarcity of CTCs, isolating enough CTCs is a crucial step for single-cell sequencing, which relies on the efficiency of enrichment and isolation of the system. With the advancement of microfluidic instruments in recent years, the enrichment and capture efficiency of most technologies can reach 80–90% or higher [12]. The CTC platforms based on microfluidic technology have better capture efficiency than other technologies. However, the microfluidic system requires a relatively slow flow rate when the blood sample passes through the chip so that the chip can fully capture the CTCs. In addition, the affinity-based method requires a long incubation time. Both requirements result in a long processing time and low throughput of the laboratory procedure. In this study, we integrated multiple steps and utilized automation to reduce time consumption and loss of CTCs. Although some systems use whole blood directly to enrich CTCs, these methods suffer from contamination of background cells [13]. The blood samples in this study still need to be pre-processed before enrichment to increase the capture efficiency and reduce the contamination of background cells. This step still needs further automation.

Currently, it remains a major challenge to precisely isolate the target cells. The process of isolating or detaching cells from the chips may result in a loss of cells. Therefore, most studies can only analyze bulks of CTCs for whole genome sequencing (WGS) or whole exome sequencing (WES) due to the technical limitation of CTC isolation [14]. In the present study, the enrichment efficiency of the Cell Reveal^TM^ system for the cell line was 81.8%. The system can accurately pick most target cells in all three patients. The results show that the system can effectively reduce the cell loss rate, and the final isolated sample has only a few contaminating background cells, which increases the success rate of single-cell sequencing.

Thyroid transcription factor 1 (TTF-1) is selectively expressed in lung and thyroid cancers and is extremely uncommon in other kind of cancers, which makes it as a useful biomarker for lung cancer [15,16]. So far, only limited studies have demonstrated a correlation between TTF-1^+^ CTCs and lung cancer. In 2016, Lu et al. reported their results by using TTF-1^+^/CK7^+^ CTCs to differentiate three patients with advanced lung cancer from patients with other cancers [17]. However, about 30% of lung adenocarcinomas are TTF-1^−^, and TTF-1 is also expressed in other cancer types [15]. Therefore, several tumor-specific markers may need to be checked when trying to differentiate pulmonary adenocarcinoma from other cancers. Another study by Messaritakis et al. also demonstrated that TTF-1^+^ CTCs could be detected in 36.4% of patients with small cell lung cancer [18]. Although the number of cases is limited, this study also shows that TTF-1^+^ CTCs may be a potential biomarker for lung cancer. However, more clinical validation is required before further conclusions can be drawn especially studies to explore the diagnostic value of TTF-1^+^ CTCs in patients whose low dose lung CT favours adenocarcinoma. Although the cut-off of TTF-1^+^ CTCs for lung adenocarcinoma remains to be determined, 0.5 TTF-1^+^ CTCs per mL of blood (2 in 4 mL) may be an appropriate value based on the data from limited cases in this study.

The phenotype of CTCs is related to tumor evolution. Therefore, single-cell sequencing on CTCs of various specific marker(s) can help us to understand tumor heterogeneity and evolution. Therefore, single-cell sequencing on CTCs of various specific marker(s) can help us to understand tumor heterogeneity and evolution. However, there is currently no report on single-cell sequencing on CTCs with specific markers to the best of our best knowledge, mainly due to technical obstacles that make it difficult to enrich enough CTCs and effectively isolate specific CTCs. This study demonstrated a high-throughput method to isolate specific CTCs for single-cell gene sequencing, which allows researchers to analyse the genetic mutation of various specific CTCs and gain a deeper understanding of tumor evolution and heterogeneity.

EGFR, KRAS, and STK11 genes are known to be mutated in lung adenocarcinoma with a high frequency as well as ERBB2 with a lower frequency. EGFR and KRAS are onco-genes and another two genes are tumor-suppressor genes [19]. All of the mutations are reported previously. EGFR p. T725M mutation is a rare and activating mutation which can increase EGFR auto-phosphorylation activity [20]. The discovery of KRAS p. R97K mutation in non-small cell lung cancer has been submitted to ClinVar database (Accession: RCV000150887.2). We discovered another mutation type R97I in the same position. STK11 exon 1–2 mutations such as potential oncogenic activity in non-squamous non-small-cell lung cancer [21]. ERBB2 exon 20 insertion is sensitive to pyrotinib and related to the therapy in patients with lung adenocarcinoma [22].

## 4. Materials and Methods

The test workflow of CTCs can be divided into the following several steps: collection and preprocessing of blood sample, enrichment and immunofluorescence staining of CTCs, as well as CTCs scanning and identification. Subsequently, target CTCs were isolated with high purity by an automatic cell picker for single-cell sequencing. Figure 5 demonstrates the laboratory workflow of the present study.

### 4.1. Cell Line and Spiking Test

The human lung adenocarcinoma cell line, H1975 (ATCC^®^ CRL-5908, Manassas, VA, USA) was used for the cell spiking test to determine the sensitivity and linearity of the method. The H1975 cells were cultured in an RPMI-1640 medium (Gibco, Grand Island, NY, USA), supplemented with 10% fetal bovine serum (FBS) and 100 units/mL penicillin (Gibco, Grand Island, NY, USA). The cultures were lifted by Trypsin/EDTA at appropriate confluency. Lifted H1975 was then incubated with CellTracker^TM^ Green CMFDA Dye (ThermoFisher, Waltham, MA, USA) for 30 min, RT. Peripheral blood mononuclear cells (PBMCs) were separated from the whole blood of healthy volunteers by a density gradient centrifugation method. Both H1975 and PBMCs were fixed by 4% paraformaldehyde for 15 min, RT. Fixed cells were then incubated with capture antibody for 30 min, 37 °C. The mixtures were prepared by different numbers of H1975 spiked into the same volume of PBMCs. The mixed cell suspension of H1975 and PBMCs were injected into the Cell Reveal^TM^ system and then the enrichment, staining, and identification steps proceeded automatically following the steps described above. In the statistical analysis, linear regression analyses were performed with GraphPad Prism 9 (GraphPad Software, Inc., San Diego, CA, USA).

### 4.2. Patient Sample Preparation

Blood samples were obtained from 3 patients with cancer (2 lung cancers and one double cancer) and 10 healthy volunteers. The blood samples were obtained after discarding the first 2 mL of blood to avoid contamination from skin epithelial cells and then loaded into BD vacutainer ACD Solution A blood collection tubes (Becton Dickinson, Franklin Lake, NJ, USA) for further processing. The diagnosis and staging of lung cancer were based on a histological examination and surgical records. The study was approved by the Ethics Committee of Taiwan Adventist Hospital. All methods were carried out in accordance with relevant guidelines and regulations, and each participant completed a written consent.

The blood samples were normalized to 2 or 4 mL for each test. Therefore, several tests could be performed simultaneously to each subject by using various tumor-associated antibodies. Whole blood samples were purified using Lymphoprep^TM^ density gradient medium (STEMCELL Technologies, Vancouver, BC, Canada) according to the supplier’s recommendation for the enrichment of the PBMC fraction.

Isolated PBMCs were fixed by 4% paraformaldehyde for 15 min at room temperature. Fixed PBMCs were then treated with an antibody cocktail containing 1:40 biotinylated anti-EpCAM antibody (R&D Systems, Minneapolis, MN, USA) and 0.05 mg/mL biotinylated anti-E-Cadherin antibody (R&D Systems, Minneapolis, MN, USA) at 37 °C, mixed consistently for 30 min. Then, 3 mL DPBS was added to the mixture of PBMCs and antibody cocktail and was centrifuged at 400× *g* for 5 min to collect the cell pallets and remove supernatant.

### 4.3. CTC Enrichment and Identification

Cell Reveal^TM^ machine (CytoAurora Biotechnologies, Inc., Hsinchu, Taiwan) was used for the CTCs enrichment and staining as described in our previous reports [23,24]. The system is a fully automatic CTC enrichment and staining system centered on an immune-affinity microfluidic chip, V-BioChip (CytoAurora Biotechnologies, Inc., Hsinchu, Taiwan) [25,26]. After placing the required reagents in the machine and setting the experiment condition and procedure, the prepared blood sample was injected into the instrument, and then the whole process would automatically proceed. The input blood sample was fixed in 4% paraformaldehyde and then mixed with 0.1% of Triton X-100 (ThermoFisher, Waltham, MA, USA) and 2% BSA (Bovine serum albumin) to increase the cellular permeability. Subsequently, the sample passed through the V-BioChip at a flow rate of 0.6 mL/h, allowing the target cells to be captured by the chip. In the present study, a mixture of anti-EpCAM antibody and anti-N-cadherin antibody was used as enrichment antibodies, while anti-TTF-1 antibody and various tumor-associated antibodies were used for immunocytochemistry staining to identify CTCs. 

After completion of cell staining, the V-BioChip was moved to a modified upright fluorescent microscope which was controlled by an automated scanning system (CytoAcqImages system, CytoAurora Biotechnologies, Inc., Hsinchu, Taiwan) for whole chip image acquisition. Cell Analysis Tools (CAT; CytoAurora Biotechnologies, Inc., Hsinchu, Taiwan) system is a tool for cell identification based on image recognition of immunofluorescence staining. The CAT system can screen the entire image within 10 min, identify the target cells and record the exact position of the target cells on the chip. CTCs were defined as intact cells with positive of tumor-associated antibody(ies)/CD45^−^/DAPI^+^ staining while TTF-1^+^ CTCs was defined as intact cells with TTF-1^+^/CD45^−^/DAPI^+^ staining. Figure 6 showed immunofluorecent staining of H1975 cells.

### 4.4. CTCs Isolation

The target cells were isolated by the Cell Picker (CytoAurora Biotechnologies, Inc., Hsinchu, Taiwan) and were then dispensed into a PCR tube for further WGA assay. The Cell Picker is a system that integrates a motorized upright fluorescence microscope and a micropipette module (Figure 7). The microscope system is equipped with a motorized linear XYZ stage, a built-in LED light source and filter sets for FITC and DAPI wavelengths. Two high-resolution cameras were used to monitor the process of CTC isolation. One vertical camera with 10X objective was used for monitoring low-light excitation fluorescence from cells and another horizontal camera (side-view camera) was used for monitoring a glass capillary tube, respectively (Figure 7b). The Cell Picker system can pick single target cells accurately and rapidly according to the target cells location information recorded by CAT, the glass capillary tube and micropipette module on the *Z*-axis, and can subsequently spit the target cells into the Eppendorf PCR tube with 4 uL Tris-EDTA buffer. 

### 4.5. Whole Genome Amplification (WGA)

DNA of CTCs were amplified by using PicoPLEX Single-Cell WGA Kit (Takara Bio, Mountain View, CA, USA). During each WGA experiment, a positive control DNA and a no template control were performed to monitor the amplification efficiency and contamination. After WGA, the DNA was purified by QIAPrep^®^ Spin Miniprep kit (Qiagen Inc., Valencia, CA, USA). The concentration and purity of purified DNA was determined by Nanodrop 2000 (ThermoFisher, Waltham, MA, USA) and size distribution was measured by the Agilent 4200 TapeStation with Genomic DNA ScreenTape assay (Agilent Technologies, Santa Clara, CA, USA).

### 4.6. PCR-Based Targeted Sequencing

Targeted sequencing was performed using the SureMASTR Tumor Hotspot kit (Agilent Technologies, Santa Clara, CA, USA). The SureMASTR technology is based on multiplex Polymerase Chain Reaction (PCR) amplification of the regions of interest. The hotspots include the following 26 cancer genes: AKT1, ALK, BRAF, CDNK2A, CTNNB1, DDR2, EGFR (ERBB1), ERBB2 (HER2), ERBB4, FGFR2, FGFR3, H3F3A, HIST1H3B, HRAS, IDH1, IDH2, KIT, KRAS, MEK1 (MAP2K1), MET, NRAS, PDGFRA, PI3KR1, PIK3CA, PTEN, and STK11. The amplified-DNA was subjected to two-step PCR reactions. The first step is to amplify the target regions, while the second is to introduce the amplified DNA with adaptor sequence. The final DNA library of each sample was diluted to a final concentration of 2 nM. A PhiX spike in the sample library pool and the PhiX control were then diluted to 10 pM. The amplicon sequencing libraries were sequenced on Illumina MiSeq (Illumina, Inc., San Diego, CA, USA) by paired-end sequencing (2 × 151 bp).

### 4.7. Next-Generation Sequencing Analysis (For Panel_File)

Sequencing adapters and 3′ low-quality bases were trimmed from raw sequencing reads using Trimmomatic (v0.36) [27]. After adapter trimming and quality control, sequencing reads were mapped to the GRCh38 Human reference genome using BWA-MEM (v0.7.17) [28]. Next, all variants (SNPs, insertions, deletions) on genes were called to target BED (SureMASTR Tumor Hotspot amplicon region) file using GATK-HaplotypeCaller (v4.1.9) [29]. Finally, functional effects of variants were annotated using Ensembl VEP (v100.0) [30]. Each variant was investigated in its potential pathogenic role with both prediction algorithms such as SIFT [31], PolyPhen-2 [32], etc., and WEB databases such as COSMIC [33], dbSNP [34], etc.

### 4.8. Next-Generation Sequencing Analysis (For Panel_with_deDup_File)

Raw sequencing FASTQ files were processed using AGeNT (v2.0.5) to remove the unique molecule identifiers UMI sequence from the beginning of each read and appended the UMI sequence to the read name in the umi. FASTQ file. Then, sequencing adapters and 3′ low quality bases were trimmed from raw sequencing reads using Trimmomatic (v0.36) [27]. After adapter trimming and quality control, sequencing reads were mapped to the GRCh38 Human reference genome using BWA-MEM (v0.7.17) [28]. Next, custom scripts were applied to remove PCR duplications based on information regarding UMI using GATK-MarkDuplicates (v4.1.9) [29] with umi.FASTQ.file. Then, variants (SNPs, insertions, deletions) on genes were called to target BED (SureMASTR Tumor Hotspot amplicon region) file using SAMtools mpileup (v1.9) [35] with default parameter and GATK-Mutect2 (v4.2.0) [29], following the best practices. Finally, the functional effects of variants were annotated using Ensembl VEP (v100.0) [30]. Each variant was investigated in its potential pathogenic role with both prediction algorithms such as SIFT [31], PolyPhen-2 [32], etc., and WEB databases such as COSMIC [33], dbSNP [34], etc.

### 4.9. Strategies of Prioritizing Variants

First, we pre-filtered variants whose MAF was less than 0.05 and mapped variants to public databases such as ClinVar [36], DisGeNET [37], and MASTERMIND [38]. Second, we collected candidate variants which were associated with lung cancer from previous results and further checked the supporting reads. Third, to include more information on variants, we not only used PubMed and PharmGKB [39] to construct therapy and drug relationships, but also used Pfam [40] and Uniprot [41] to acquire protein domain and functional information from pre-filter variants. Fourth, we used MASTERMIND [38] to discover other candidate variants which may have been related to lung cancer and sequencing reads that were larger than 100 from pre-filter variants. Finally, we combined the above results to decide the importance of variants. Although those mutations are rare mutations, there are some mutations that have very high variant allele fraction. For example, DDR2 p.G505S mutation, whose responses to dasatinib has high variant allele fraction because there is only one alteration read in the position. We removed these mutations from the results because it may have been affected by the sequencing condition.

## 5. Conclusions

In this study, we integrated an immunoaffinity based microfluidic device, an AI-assisted image recognition system and a robotic cell picking device to isolate specific CTCs from the peripheral blood of three cancer patients, and to successfully detected cancer related gene mutations by using commercially available genetic testing instruments. The study also demonstrated TTF-1^+^ CTCs as a potential biomarker. It should be highlighted that the entire experimental process of CTCs enrichment, staining, identification, and isolation has been almost fully automated, which brings the advantages of low cost, high throughput, time saving, labor saving, and high performance. Although more clinical validation is needed, I believe our experience is very helpful for single-cell sequencing and even multi-omics analysis of CTCs. 

## Figures and Tables

**Figure 1 ijms-23-15139-f001:**
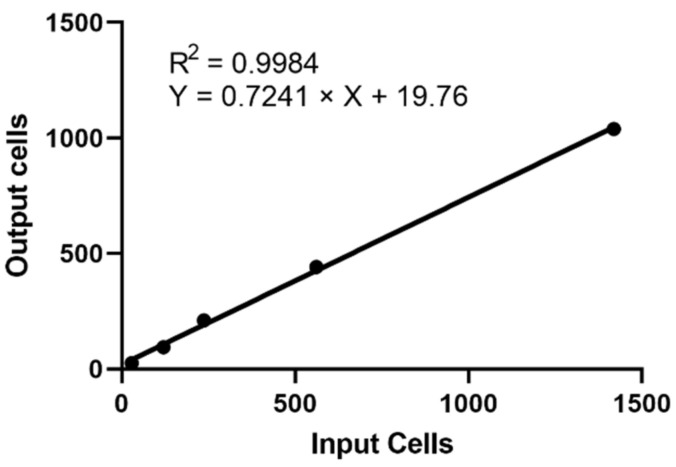
Linearity of tumor-derived cell line detection in spiking experiments. The experiment showed a recovery (mean 81.8%) and a linear performance (R^2^ = 0.998) of the protocol. Known numbers of H1975 were spiked into healthy donor PBMCs. The numbers of spiked and detected H1975 cells are plotted on the x- and y-axis, respectively.

**Figure 2 ijms-23-15139-f002:**
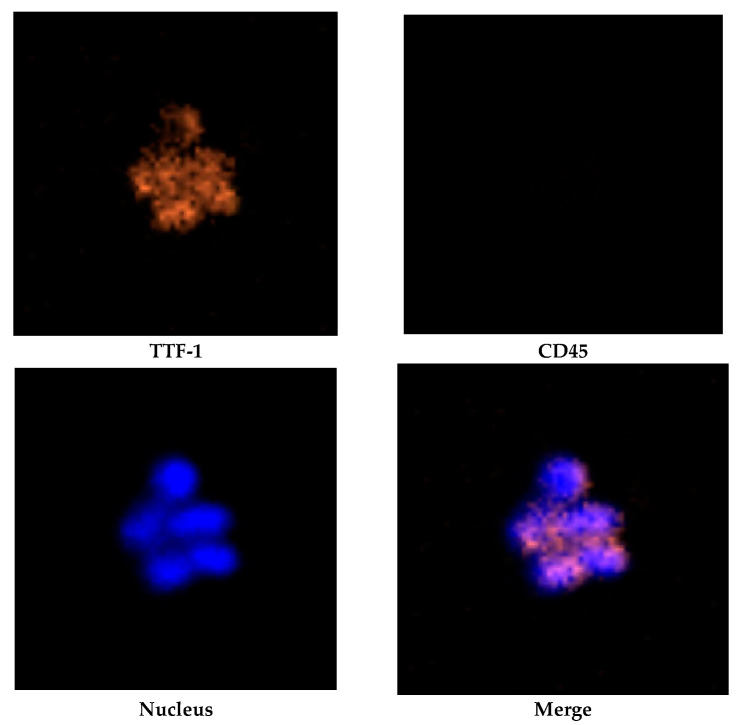
Cluster of TTF-1^+^ CTCs in case 3 with immunofluorescent staining of TTF-1^+^/CD45^−^/DAPI^+^.

**Figure 3 ijms-23-15139-f003:**
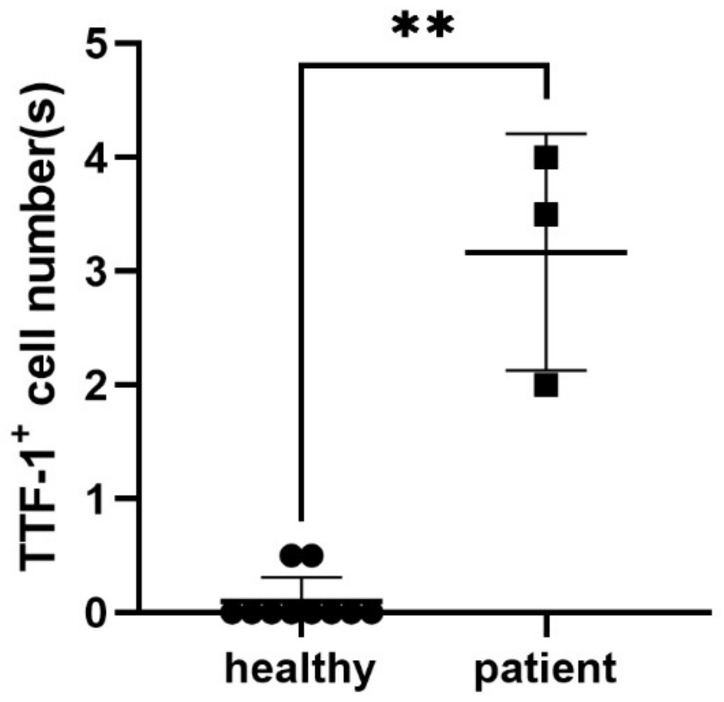
Detection of TTF1^+^ CTCs in healthy donors and lung cancer patients. The difference of TTF-1^+^ CTCs between healthy donors (N = 10) and patients with cancer (*n* = 3) was statistically significant. **, *p* < 0.05, by Mann-Whitney test.

**Figure 4 ijms-23-15139-f004:**
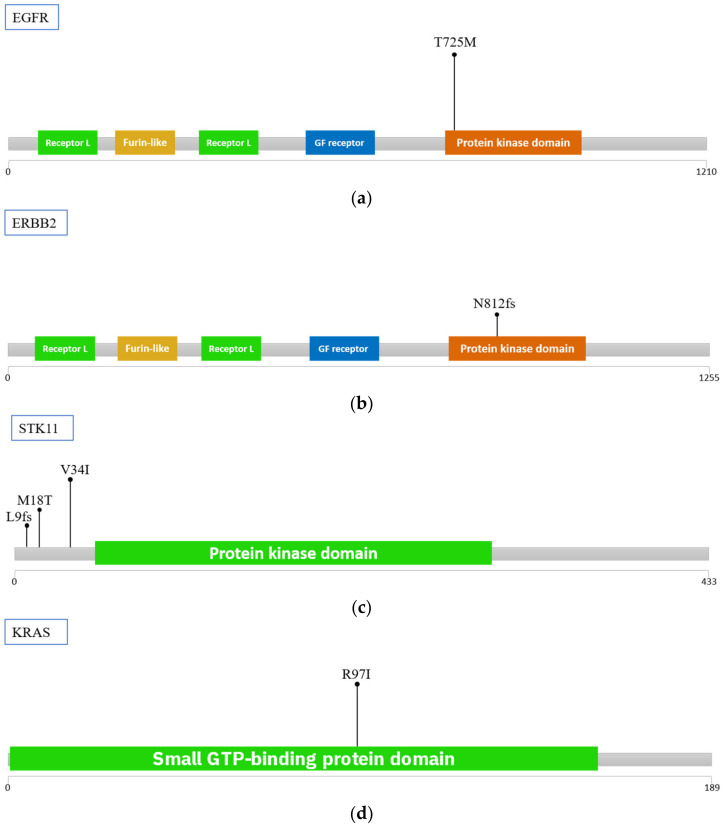
Mutations distribution in functional domain (**a**) EGFR (**b**) ERBB2 (**c**) MET (**d**) STK11. Most of mutations on protein kinase domain.

**Figure 5 ijms-23-15139-f005:**
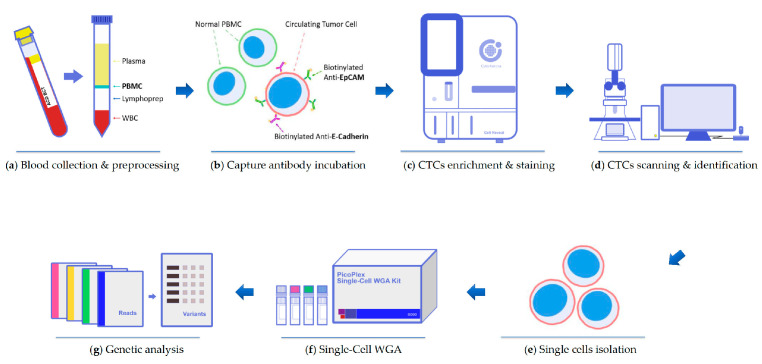
Schematic of workflow of circulating tumor cells (CTCs) enrichment and characterization. (**a**) Density gradient centrifugation was used to isolate PBMCs from the blood sample. (**b**) PBMCs were incubated with by biotinylated antibodies. (**c**) CTCs were enriched and stained through the Cell Reveal^TM^ machine. (**d**) The whole chip image was acquired by an automatic scanning system that was controlled by CytoAcqImages software Cell Analysis Tools (CAT) can identify the target cells record the position and the morphology of CTCs. (**e**–**g**) High purity single cells were isolated by using Cell Picker for whole genome amplification and genome analysis.

**Figure 6 ijms-23-15139-f006:**
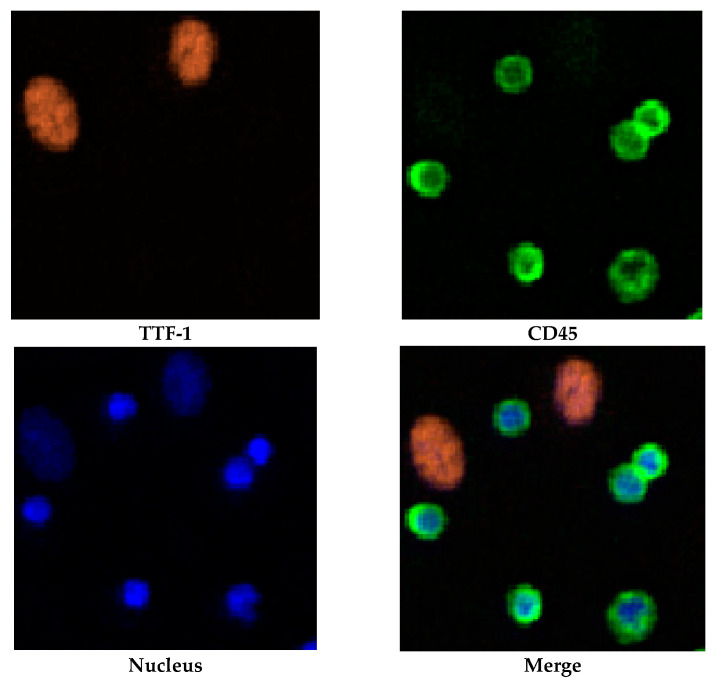
Immunofluorescent staining of H1975 and PBMCs. TTF-1^+^ CTCS showed immunofluorescent staining of TTF-1^+^/CD45^−^/DAPI^+^ while PBMCs showed TTF-1^−^/CD45^+^/DAPI^+^, respectively.

**Figure 7 ijms-23-15139-f007:**
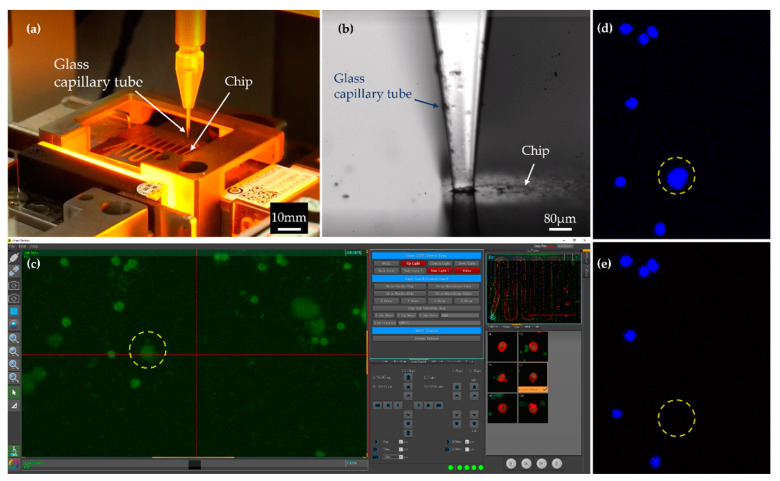
(**a**) Internal image of Cell Picker. (**b**) The glass capillary tube touches the chip surface and precisely frames the target cell. (**c**) Image of control interface of Cell Picker. The cells are monitored under a microscope with a 10× objective. (**d**,**e**) An example of picking images, before and after images acquired at 10× showing the target cell was isolated from a chip. The nuclei are stained blue.

**Table 1 ijms-23-15139-t001:** Clinical diagnosis, status and CTC counts of the three patients.

Case No.	1	2	3
Sex	F	F	M
Age	62	49	67
Diagnosis/Staging	Lung AdCa/I	Lung AdCa/IV	Double cancer (lung AdCa/I & thyroid papillary carcinoma/IV)
Status	Post-operation	Post-treatment	Post-operation
CTCs			
TTF-1^+^ CTCs	14/4 mL (2 mL × 2)	8/2 mL	16^#^/8 mL (4 mL × 2)
TTF-1^−^ CTCs	18/4 mL (2 mL × 2)	1/2 mL	12/8 mL (4 mL × 2)
Total CTC count	32/4 mL (2 mL × 2)	9/2 mL	28^#^/8 mL (4 mL × 2)
Isolated CTCs			
TTF-1^+^ CTC	6	8	10
TTF-1^−^ CTC	6	0	0
WBCs	10	2	17
Purity (%)	55	80	37

AdCa: adenocarcinoma; ^#^ including 12 cells in 3 clusters.

**Table 2 ijms-23-15139-t002:** All samples depth and On-target percentage.

Case No	Average Depth	On-Target Percentage (10×)	On-Target Percentage (100×)	On-Target Percentage (250×)
1	7560	89%	84%	79%
2	8014	62%	57%	54%
3	5852	90%	79%	71%

**Table 3 ijms-23-15139-t003:** Mutations detected in CTCs of the three patients.

Gene	Variation	Patient	CTCs AF	Pathway
EGFR	p.T725M	STE504366	0.235	1. MAPK_SIGNALING_PATHWAY 2. ERBB_SIGNALING_PATHWAY 3. PATHWAYS_IN_CANCER
ERBB2	p.N812fs	RCE162277	0.087	1. ERBB_SIGNALING_PATHWAY 2. PATHWAYS_IN_CANCER
ERBB2		RCE385327	0.127	
KRAS	p.R97I	RCE162277	0.253	1. ERBB_SIGNALING_PATHWAY 2. PATHWAYS_IN_CANCER
STK11	p.L9fs	STE504366	0.2	1. MTOR_SIGNALING_PATHWAY 2. ADIPOCYTOKINE_SIGNALING_PATHWAY
STK11	p.M18T	STE504366	0.267	
STK11	p.V34I	STE504366	0.275	

AF: allele frequency.

## Data Availability

Data are available upon request.

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
