# Peer review of "Isolation of TTF-1 Positive Circulating Tumor Cells for Single-Cell Sequencing by Using an Automatic Platform Based on Microfluidic Devices"

_ijms, 2022, doi:10.3390/ijms232315139_

Round 1
Reviewer 1 Report
GOOD EFFORT.THE PROBLEM, I CONSIDER AS MOST IMPORTANT ,IS THE >30% TTF-1 NEGATIVE ADENOCARCINOMAS.WHAT HAPPENS WITH THEM?MOREOVER TTF-1 IS POSITIVE IN OTHER CANCERS(EVEN IN MINOR %).THE IDEA IS GOOD,THE METHODOLOGY IS ADEQUATE BUT YOU SHOULD PROPOSE SOMETHING FOR THESE CASES,FOR EXAMPLE CREATE A PROPOSAL OF GUIDELINES :EXAM ONLY TTF-1 POS.PATIENTS,CT SCAN FAVOR ADENOCARCINOMA OF THE LUNG,CUT-OFF IN CELL NUMBER,ETC.
Reviewer 2 Report
This manuscript reported a targeted strategy for circulating tumor cell capture via a specific biomarker and its antibody with help of microfluidic chip, accompanied by automatically identification of the captured CTCs through immunostaining technique. And some representative gene mutations in the CTCs mainly derived from lung cancer patients was analyzed by using single-cell sequencing technique. A sophisticated methodology was involved in this study by combining biomarker for CTCs targeting, microfluidic chip, automatic machine for identification, and single cell gene sequencing. It is a somewhat interesting work, however, the scientific significance and novelty of this manuscript was greatly reduced considering following concerns. So at this stage, it is difficult to be accepted for publication in this journal.
1. Despite using integrated methods to enrich and identify CTCs and also perform gene sequencing, a clear scientific intention and hypothesis was missed.
2. The authors have repeatedly emphasized one feature of the study on automatic instrument for CTC capture and identification, but in the experimental process, it is necessary to conduct gradient centrifugation on the clinical patient's blood sample to collect monocyte layer, then bind to the cells with the antibody off the chip, and then add them to the microfluidic chip for subsequent operations. Can the authors directly use whole blood for corresponding operations? If not, what is the significance and value of using microfluidic chips?
3. For tumor precision medicine based on CTC single cell sequencing, one of the important concerns is the heterogeneity of cells and the representativeness of captured cells. How to ensure the representativeness of cells when using one kind of specific antibody conjugated interfaces to capture and separate CTC? When combined with single cell sequencing for detection and analysis of cancer occurrence, development and prognosis, whether target cell identified by a specific biomarker highlighted in this paper is a strength or a weakness?
4. There is a problem in Figure 6. The location of CTC and blood cells in the graph of merge is inconsistent with the fluorescent images; In Fig. 7a and 7b, the scale bars are need to be added.
Round 2
Reviewer 2 Report
The authos have addressed my concerns and improved legibility of the manuscript. I suggest to accept the manuscript for publication in this journal.